# Pathogen-Reservoir Interactions: What We Do Not Know Likely Will Hurt Us

**DOI:** 10.3390/v13020195

**Published:** 2021-01-28

**Authors:** Charles H. Calisher

**Affiliations:** Arthropod-Borne and Infectious Diseases Laboratory, Department of Microbiology, Immunology and Pathology, College of Veterinary Medicine and Biological Sciences, 3195 Rampart Rd., Foothills Campus, Colorado State University, Fort Collins, CO 80523-1690, USA; calisher@cybersafe.net

**Keywords:** SARS-CoV-2, reverse transmission, coronavirus, bats, zoonosis, reservoir, experimental infections, anthroponosis, pathogenesis

## Abstract

The establishment of selective colonies of potential vertebrate hosts for viruses would provide experimental models for the understanding of pathogen-host interactions. This paper briefly surveys the reasons to conduct such studies and how the results might provide information that could be applied to disease prevention activities.

The majority of current methods used to detect viruses in clinical and field-collected materials, including tissues from patients, wild animals, and livestock, are based on various iterations of polymerase chain reactions. These can provide highly specific information regarding the uniqueness of nucleotide sequences of literally thousands of viruses. Distinct sequences, even if they are not exact matches with recognized viruses, may be considered close enough to previously known sequences to suggest or indicate relationships between these viruses. For example, the consensus terminal nucleotide sequences of the large (L), medium (M), and small (S) genome segments of viruses of the family *Orthobunyaviridae* (orthobunyaviruses), genus *Orthobunyavirus*, are UCAUCACAUG at the 3′ end and AGUAGUGUGC at the 5′ end. If those sequences are found, the virus is an orthobunyavirus, and if not, it is not a member of that genus.

However, genetic sequences do not nearly tell us everything we need to know about viruses. Extrapolating features, traits, or behaviors from sequences of well-characterized viruses to those of recently discovered virus sequences are presumptions, adequate for generalizing but neither exact nor precise, and certainly not definitive because biological characteristics can only be determined from biological characterizations. For example, the recently recognized pandemic coronavirus (phylum *Pisuviricota*, class *Pisoniviricetes*, order *Nidovirales*, family *Coronaviridae*, subfamily *Orthocoronavirinae*, genus *Betacoronavirus*, species *Severe acute respiratory syndrome coronavirus 2*), SARS-CoV-2, is similar to its closest relative (SARS-CoV-1) in many ways and differs from SARS-CoV-1 in many other ways. For example, in comparison with SARS-CoV-1 infections of humans, SARS-CoV-2 has a somewhat longer incubation period, infections result in a greater proportion of mild illnesses, require less hospitalization and intensive care, and result in infections that are highly infectious on the day of onset and maximum level of infectivity. SARS-CoV-2 is much more contagious than SARS-CoV-1 but is less pathogenic and with lower case-fatality rates than does SARS-CoV-1. Numerous other differences between the two viruses have been observed. Needless to say, nucleotide sequence analyses, such as next-generation sequencing, have provided fascinating and useful genetic data but have not provided definitive biological information that would allow us to further understand and treat or prevent this pandemic disease or other diseases for which such information is lacking, thus is not at the present routinely depended on in clinical practice.

SARS-CoV-2 has been shown to infect a wide variety of mammals, including gorillas (*Troglodytes gorilla*), house cats (*Felis catus*), lions (*Panthera leo*), tigers (*Panthera tigris*), and snow leopards (*Panthera uncia*), dogs (*Canis familiaris*), American minks (*Neovison vison*), deer mice (*Peromyscus maniculatus*), Sunda pangolins (*Manis javanica*), ferrets (*Mustela putorius furo*), and bats (Chiroptera) of numerous species [1,2,3,4]. As far as this author knows, this virus does not cause fatalities in any of these vertebrates, with the exception of minks [5]. However, some of these mammals might very well serve as reservoirs of SARS-CoV-2; some commercially raised mink in North America have escaped captivity and a few of those that have been recaptured were shown to be infected, so we might soon learn whether they do or do not serve as virus reservoirs. Thus, reverse transmission, that is, anthroponosis (infected humans to uninfected domesticated or wild animals), is possible and the impact on subsequent persistence of SARS-CoV-2 is enormous and alarming. Dogs and pigs appear to be susceptible to SARS-CoV-2, while vertebrates of other domesticated species, including poultry, do not [6].

One may wonder how a virus “jumps” from an infected member of one species to an uninfected member of a different species. Simple exposure and the resulting subsequent infection are the answer. The mechanism by which a new virus arises is more complex. Among the many possible scenarios for an initial transmission event might be pleiotropism, that is, a viral gene causing one effect in the originating host and a different effect in a subsequent host. It can easily be imagined that when two closely related viruses co-infect a vertebrate or invertebrate, host recombination of viral genomes might occur. For example, a coronavirus infecting an asymptomatic but coronavirus-infected pangolin might recombine with a different, perhaps also an otherwise harmless, coronavirus resulting in a pathogenic viral recombinant. Should that virus infect a disease-resistant bat, which subsequently transmits it to other bats, virus persistence might follow. Alternatively, if the recombinant is transmitted to a disease-susceptible human, such as SARS-CoV-2, a pandemic might ensue. That another step (transmission to yet another host) might occur in such a consequence is relatively trivial. Zoonotic diseases (transmitted from non-human animals to humans) and anthroponotic diseases (transmitted from humans to other animals) are essentially two sides of the same coin.

It is likely, but not proven, that the first transmission of SARS-CoV-2 to humans was a coronavirus of bats that somehow infected an intermediate vertebrate host, which then served as the virus source for the initial infection of a person who then served as the primary human infection of this pandemic. Over the past two decades, it has become clear that bats are reservoir hosts of many viruses, perhaps 200 or many more by now, and SARS-CoV-1 is recognized as a “bat virus” [7].

There are 39 species of coronaviruses recognized at this time. Recombination between them has been shown by direct RNA genomic analyses of viruses from nature and by laboratory experimentation and analyses. The two viruses causing SARS and Middle East respiratory syndrome coronavirus, which are closely related to SARS-CoV-2, may have arisen from recombination events that occurred in the distant or recent past. Given the millions of COVID-19 cases worldwide and the obvious potential for further zoonotic and anthroponotic viral transmission, further research and surveillance activities are needed to definitively determine the role of animals in community transmission of SARS-CoV-2 and many other viruses. In that case, why have there not been large-scale, well-funded, and organized efforts to conduct prospective pathogen/reservoir interactions so that we already have in hand useful data when the next pandemic or severe epidemic is recognized?

Given that there are more than 60,000 species of vertebrates (mammals 6515 [8]; birds 10,425; reptiles 10,038; amphibians 7302; and fishes 32,900), and that rodents and bats alone comprise 2590 and 1432 species, respectively, it would be impossibly expensive and logistically hopeless to attempt or even consider establishing pathogen-reservoir interactions for all known viruses, principally because there are thousands of recognized viruses of vertebrates alone. In addition, a wide variety of mammals would be needed for different geographic areas where these thousands of viruses are known to occur in nature. Nonetheless, reservoir-specific immune reagents might be prepared and made available by organizations such as the American Type Culture Collection; this would be at least partially helpful.

To begin, were we to have available colonies of obviously relevant species we could begin potential pathogen-reservoir studies that could serve as predictors of what we might expect should an outbreak of those pathogens be recognized. Having such information in a complete database is likely only a dream, but we could, at least, expand what has already been done for decades—establish colonies of vertebrates of recognized target species and conduct experimental infections with them. Certain colonies of such vertebrates already have been established for this purpose. Such facilities as are already being used are equipped at appropriate safety levels, essentially all at biosafety level 4, with well-trained investigators, those who themselves have been trained to handle hazardous materials including, of course, hazardous pathogens. In such a way, we might accumulate information that would contribute to our knowledge of the potential pathogenesis of viruses and their pathogenetic mechanisms. Saving time would almost assuredly save lives.

## Data Availability

Not applicable.

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
