# Peer review of "Pathogen-Reservoir Interactions: What We Do Not Know Likely Will Hurt Us"

_viruses, 2021, doi:10.3390/v13020195_

Round 1
Reviewer 1 Report
The article is a brief Perspective on the merits and challenges of establishing "selective colonies of potential vertebrate hosts for viruses".
Early on, the author remarks that genotyping efforts "based on various iterations of polymerase chain reactions" are inadequately informative as to the nuances of virus biology, comparing and contrasting SARS-CoV-1 and SARS-CoV-2 sequences and behaviors as an example. A brief discussion of the likely origins and broad interspecies tropism of SARS-CoV-2 follows, after which a summary of the species diversities of coronaviruses and other viruses is presented. The author concludes by arguing, in essence, that although it is "hopeless to attempt or even consider to establish pathogen-reservoir interactions for all known viruses", by expanding and studying existing "colonies of vertebrates of recognized target species" we might "accumulate information that would contribute to our knowledge of the potential pathogenesis of viruses and of their pathogenetic mechanisms".
The merits of studies such as those advocated here are beyond dispute; however, a less one-sided assessment of the advantages and disadvantages of genotyping vs. establishing "vertebrate colonies" seems warranted. In particular, as the author readily acknowledges, the capacity of the latter approach for broad analysis of the myriad virus species is grossly inadequate. By contrast, the throughput of viral genetic analyses using next generation sequencing techniques is vastly higher. Hence, revising this Perspective to present the pros and cons of the two approaches in a more balanced manner would be welcome.
Finally, the manuscript is well written and clearly presented. Only a few typographical errors of "SARS-C0V-2" (vs "SARS-CoV-2") need to be corrected.
Author Response
Typographic errors (SARS-CoV-2) now corrected and unitalicized species names now italicized.
Expansion of contrast between genetic and biological methods of analyzing potential viral pathogenetic characteristics was expanded (lines 46-49) only somewhat. The author did not originally intend to go into such detail but added a sentence or two in this regard.
Acknowledgment added.
Reviewer 2 Report
This paper is interesting, well written, and deals with a currently "hot" issue. However, in spite of an excellent introduction, the matter of interest appears only in the last paragraph, as if the authors "ran out of steam". I would broaden the discussion in one of two ways: Compare the COVID-19 situation with the recent influenza epidemic, elaborating on the role of pigs and ducks. Another option is to describe the suggested facility, its logistics and safety issues, as well as research methodologies that should be used.
Author Response
Added a bit regarding facilities, personnel, and safety issues required for the work suggested (lines 114-118).